# APOBEC3B Is Co-Expressed with PKCα/NF-κB in Oral and Oropharyngeal Squamous Cell Carcinomas

**DOI:** 10.3390/diagnostics13030569

**Published:** 2023-02-03

**Authors:** Galinos Fanourakis, Efthymios Kyrodimos, Vasileios Papanikolaou, Aristeidis Chrysovergis, Georgia Kafiri, Nikolaos Papanikolaou, Mihalis Verykokakis, Konstantinos Tosios, Heleni Vastardis

**Affiliations:** 1Department of Oral Biology, School of Dentistry, National and Kapodistrian University of Athens, 2 Thivon Str., 11527 Athens, Greece; 21st ENT Department, Hippokration Hospital, School of Medicine, National and Kapodistrian University of Athens, 114 Vasilissis Sophias Ave., 11527 Athens, Greece; 3Department of Pathology, Hippokration Hospital, 114 Vasilissis Sophias Ave., 11527 Athens, Greece; 4EnzyQuest PC, Science and Technology Park of Crete, 100 Nikolaou Plastira Str., Vassilika Vouton, 70013 Heraklion, Greece; 5Institute for Fundamental Biomedical Research, BSRC Alexander Fleming, 34 Fleming Str., 16672 Vari, Greece; 6Department of Oral Pathology, Medicine and Hospital Dentistry, School of Dentistry, National and Kapodistrian University of Athens, 2 Thivon Str., 11527 Athens, Greece; 7Department of Orthodontics, School of Dentistry, National and Kapodistrian University of Athens, 2 Thivon Str., 11527 Athens, Greece

**Keywords:** APOBEC3B, PKCα, NF-κΒ, head and neck squamous cell carcinoma

## Abstract

The enzymatic activity of APOBEC3B (A3B) has been implicated as a prime source of mutagenesis in head and neck squamous cell carcinoma (HNSCC). The expression of Protein Kinase C α (PKCα) and Nuclear Factor-κΒ p65 (NF-κΒ p65) has been linked to the activation of the classical and the non-canonical NF-κB signaling pathways, respectively, both of which have been shown to lead to the upregulation of A3B. Accordingly, the aim of the present study was to evaluate the expression of PKCα, NF-κΒ p65 and A3B in non-HPV related oral and oropharyngeal squamous cell carcinomas (SCC), by means of immunohistochemistry and in silico methods. PKCα was expressed in 29/36 (80%) cases of oral and oropharyngeal SCCs, with 25 (69%) cases showing a PKCα+/A3B+ phenotype and only 6/36 (17%) cases showing a PKCα-/A3B+ phenotype. Εxpression of NF-κB p65 was seen in 33/35 (94%) cases of oral and oropharyngeal SCCs, with 30/35 (86%) cases showing an NF-κB p65+/A3B+ phenotype and only 2/35 (6%) cases showing an NF-κB p65-/A3B+ phenotype. In addition, mRNA expression analysis, using the UALCAN database, revealed strong expression of all three genes. These findings indicate that the expression of A3B is associated with PKCα/NF-κB p65 expression and suggest a potential role for the PKC/NF-κB signaling pathway in the development of oral and oropharyngeal cancer.

## 1. Introduction

Oral and oropharyngeal cancer constitutes the sixth most common malignancy worldwide [1], with squamous cell carcinoma (SSC) representing more than 90% of all oral cancers [1,2]. Smoking and excess alcohol consumption are the most significant etiological factors, acting independently or synergistically [3]. The five-year survival rate is relatively low—approximately 50%—largely due to delayed recognition and diagnosis at advanced disease stages [4].

The Apolipoprotein B mRNA Editing enzyme Catalytic polypeptide-like 3 (APOBEC3) family of enzymes is a major component of innate immunity. APOBEC3 enzymes catalyze the deamination of cytosine bases in nucleic acids and convert cytosines to uracils, thus contributing to significant mutation and damage of target DNA or RNA sequences [5,6]. Among the seven members of the human APOBEC3 family, the enzymatic activity of APOBEC3B (A3B) has been implicated as a prime source of mutagenesis in multiple human cancers [7,8,9,10,11,12]. While several studies have investigated the downstream consequences of A3B overexpression and activity, little is known regarding the upstream mechanisms of A3B dysregulation [13]. A proposed mechanism suggested that Protein Kinase C (PKC)/Nuclear Factor-kappa Β (NF-κΒ) signaling induces A3B expression [14].

PKC is a family of lipid-sensitive enzymes that has been implicated in a variety of cellular functions, including proliferation, differentiation, motility, and apoptosis. Human cells may express up to nine different *PKC* genes [15,16] that are divided into three subfamilies, i.e., conventional or classical, novel, and atypical PKC isoforms [17]. PKCα is a member of classical PKCs that is activated both by diacylglycerol (DAG) and increased intracellular concentration of Ca^2+^ [18]. Upon stimulation of PKCα with phorbol-myristate acetate (PMA), which is a DAG analogue [19,20], the noncanonical NF-κΒ/RELB signaling pathway was activated, which resulted in the upregulation of A3B [14] and modulated cell apoptosis in urothelial cell carcinoma of the bladder [21]. Interestingly, however, in another study, PKC induction by PMA caused upregulation of A3B through the classical NF-κΒ/RELA pathway [22].

NF-κΒ, a family of transcription factors, have broad roles in the regulation of gene expression in diverse cellular responses, such as immunity and inflammation [23,24,25], and in the oncogenic processes in specific malignancies [25,26,27]. In head and neck squamous cell carcinomas (HNSCC), expression and activity of NF-κΒ is often upregulated and protein levels increase gradually from premalignant lesions to invasive cancer [28,29,30]. The *A3B* promoter region includes several NF-κΒ binding sites, thus providing strong evidence for a direct linkage between NF-κΒ-mediated regulation of A3B expression [14,22,31].

Based on these findings, as well as the established role of A3B activity in HNSCC, we investigated the immunohistochemical expression of the PKCα, NF-κΒ, and A3B proteins in a cohort of non-HPV related human oral SSC (OSSC) and oropharyngeal SCC (OPSCC). In addition, we examined the mRNA expression level of *PRKCA*, *RELA* and *APOBEC3B* genes in HNSCC samples, derived from the UALCAN database.

## 2. Materials and Methods

### 2.1. Tissue Specimens and Patient Information

Archival formalin-fixed and paraffin-embedded (FFPE) specimens from 36 patients with OSCC (*n* = 18) and OPSCC (*n* = 18) were retrospectively retrieved from the files of the Department of Pathology, Hippokration Hospital, National and Kapodistrian University of Athens. All tumors were HPV-negative and graded according to the World Health Organization (WHO) classification. Data regarding age, sex, stage, and survival were available from medical records (Table 1). All samples were initially evaluated for HPV presence with immunohistochemistry, using a p16 antibody. Further evaluation and confirmation of the original analysis was assessed with DNA in situ hybridization (ISH). Hematoxylin- and eosin-stained slides were reviewed to confirm diagnosis.

### 2.2. Immunohistochemistry

Immunohistochemical staining for PKCα, NF-κB p65, and A3B was performed on 5μm-thick FFPE tissue sections. Primary antibodies employed were the mouse monoclonal PKCα (# sc-8393, Santa Cruz Biotechnologies, Dallas, TX, USA), the mouse monoclonal NF-κB p65 (# sc-8008, Santa Cruz Biotechnologies, TX, USA), and the rabbit polyclonal anti-APOBEC3B (# ab191695, Abcam, Cambridge, MA, USA). Primary antibodies were applied overnight at dilutions of 1:100, 1:75, and 1:200, respectively. PT Link pre-treatment module (DAKO, Agilent Technologies, Santa Clara, CA, USA) was used for deparaffinization, rehydration and epitope retrieval, while EnVision (DAKO, Glostrup, Denmark), a standard two-step visualization system, was implemented for all immunohistochemical procedure.

Specificity of the immunohistochemistry was demonstrated by the complete absence of staining product when omitting primary antibodies, while breast carcinoma FFPE samples were used as positive controls.

### 2.3. Evaluation of Immunohistochemistry

An investigator blinded to the origin of the FFPE samples evaluated the extent and intensity of immunoreactivity by applying the three-scale semi-quantitative system used in our previous study [11]. Extent of immunoreactivity was scored according to the percentage of stained cells as 0, <5%; 1, 5–50%; and 2, >50%. Intensity of immunoreactivity was scored according to the color of stained cells as 0, no staining; 1, yellow color; and 2, dark color. The total immunohistochemistry score (IS) was calculated using the formula IS = extent × intensity, and qualitatively characterized as no staining (IS = 0), weak-moderate (IS = 1 or 2), and intense (IS = 4) staining.

### 2.4. Data Retrieval and Gene Expression Analysis

The UALCAN portal (http://ualcan.path.uab.edu) [32] is an interactive web resource that allows researchers to access, analyze, visualize, and interpret TCGA gene expression data from 31 cancer types with several clinicopathological features, including sex, age, tumor grade, individual cancer stage, HPV status and TP53 status. Transcription data from HNSCC samples were retrieved (access date on 15 October 2022) and used to explore the expression of *PRKCA*, *RELA*, and *A3B* mRNA.

### 2.5. Statistical Analysis

For each of the PKCα, NF-κB, and A3B proteins, results were compared to assess statistically significant differences between means for sex, disease stage, and survival. For all comparisons a two-sample *t*-test assuming unequal variances was performed, assuming statistical significance when *p* < 0.05.

## 3. Results

### 3.1. PKCa Expression in Oral and Oropharyngeal SCCs

Cytoplasmic PKCα expression was observed in 29 out of 36 carcinomas (80%) (Figure 1) and scored as intense in 10 (28%) cases and weak-moderate in 19 (53%) cases (Table 2). Statistically significant positive correlations were observed for stage I vs. IV tumors, and stage II vs. III and IV tumors (Figure 2), suggesting that PKCα expression was increased in higher cancer stages. Evaluation of *PRKCA* mRNA expression in HNSCC subgroups, based on clinicopathological features in the UALCAN database, revealed positive correlation for stage I vs. II, stage I vs. IV, and stage III vs. IV (Figure 3).

### 3.2. NF-κΒ Expression in Oral and Oropharyngeal SCCs

Cytoplasmic NF-κΒ expression was positive in 33 out of 35 carcinomas (94%) examined (Figure 1) and scored as intense in 22 (63%) cases and weak-moderate in 11 (31%) cases (Table 2). Statistically significant positive correlations were observed for groups of stage I vs. IV, stage II vs. IV, pooled stage I + II vs. III + IV, and male vs. female patients (Figure 2). Analysis for the differential *RELA* gene expression in HNSCC samples from the UALCAN database showed statistically significant difference only between tumor stages I and IV, but no positive correlation between male and female patients (Figure 3).

### 3.3. A3B Expression in Oral and Oropharyngeal SCCs

A3B protein expression was detected in 31 carcinomas (86%); A3B was nuclear in two cases, nuclear and cytoplasmic in 20 cases, and cytoplasmic in nine cases (Figure 1). It was scored as intense in 18 (50%) cases and weak or moderate in 13 (36%) cases (Table 2). No statistically significant correlation was found between subgroups (Figure 2). When HNSCC subgroups from UALCAN were examined, a statistically significant correlation was observed for stage I vs. III, and male vs. female patients (Figure 3).

## 4. Discussion

In view of the fact that A3B is expressed in HNSCC [11,33] and in other cancers [7,9,34], we hypothesized that it may be involved in cancer pathogenesis. Consistent with that, patients show a significant tendency for TCW-to-TTW trinucleotide mutations, which is the preferred site of A3B activity [7,9,11]. In the present study, we report that A3B expression was associated with PKCα and NF-kB expression in non-HPV related oral and oropharyngeal SCC.

Several mutations attributable to members of the APOBEC family of proteins are found in almost all HPV-positive and many HPV-negative HNSCC [7,35,36,37,38]. Argyris et al. [39] reported high levels of A3B protein both in HPV-negative OSCC and HPV-positive OPSCC, and speculated the existence of nonviral mechanisms, in addition to HPV-associated mechanisms, where HPV proteins E6/E7 and polyoma virus large T antigen (TAg) regulate A3B expression. Recent studies have shown upregulation of A3B through activation of PKC, both through the classical NF-κB signaling transduction pathway that triggers nuclear translocation of the predominant p65(RELA)/p50(NF-κΒ1) heterodimer [22], and the noncanonical/alternative NF-κΒ pathway, which involves recruitment of RELB/p52(NF-κΒ2) [14].

NF-κΒ may promote survival in cancer cells and inflammation in the tumor microenvironment, and its expression has been associated with poor prognosis in most solid tumors [40]. In tumor cell lines, activation of both the classical [22] and the noncanonical NF-κΒ pathways [14] has been shown. In OSCC, NF-κΒ p65 involvement in the activation of the classical NF-κΒ pathway [22] has been studied extensively. NF-κΒ p65 cytoplasmic immunostaining gradually increased from normal mucosa to OSCC, indicating a role for NF-κΒ activation in the early stages of oral carcinogenesis [41], while NF-κΒ p65 expression, as determined by semiquantitative RT-PCR, immunohistochemistry, Western blot and ChIP analysis, was associated with OSCC progression and chemoradiation resistance, possibly through regulation of the antiapoptotic molecule B cell lymphoma-2 (BCL-2) [42]. A role for the Toll-like Receptor 4 (TLR4)/NF-κB p65 pathway in the malignant transformation of oral mucosa and OSCC progression was proposed based on the parallel expression of NF-κB p65 with TLR4 [43]. In the present study, cytoplasmic expression of NF-κB p65 was observed in 33/35 cases of oral and oropharyngeal SCCs examined, indicating activation of the classical NF-κB pathway; 28/35 cases showed an NF-κB p65+/A3B+ phenotype and only 2/35 cases were NF-κB p65-/A3B+.

Although PKCα activates the noncanonical NF-κΒ pathway [14], PKCα expression has not been thoroughly investigated in oral and oropharyngeal carcinomas. In two human cell lines and tissue from tongue SCCs, the mRNA levels of PKCα were moderately lower to higher in well-differentiated cells and low in poorly differentiated cells, compared to healthy controls [44]. In other studies, PKCα and other classical PKCs were shown by Western blot to be increased in tissue extracts from 29 human primary OSCCs, regardless of differentiation grade [45]. In addition, the PKCα-selective inhibitor Safingol exhibited dose-dependent growth inhibition of human OSCC cells [46,47] through endonuclease G induced apoptosis [48]. Here, we showed that PKCα was expressed in 29/36 cases of oral and oropharyngeal SCCs examined; 25 cases showed a PKCα+/A3B+ phenotype, while only 6/36 cases were PKCα-/A3B+.

In the present study, 29/31 A3B+ cases of oral and oropharyngeal carcinoma showed cytoplasmic staining, (9/31 only cytoplasmic and 20/31 both cytoplasmic and nuclear). These findings agree with previous studies in breast [49] and ovarian carcinomas [50], esophageal squamous cell carcinomas [51], and gastroenteropancreatic neuroendocrine tumors [52], where a rabbit polyclonal antibody that recognizes an epitope of 14 amino acids near the N-terminal of A3B was used, as well as with our previous work in HNSCC samples, where a rabbit polyclonal antibody raised against residues 1–100 of the human A3B protein was applied. Cytoplasmic positivity was also reported with a rabbit monoclonal antibody that recognizes a C-terminal epitope shared by A3A and A3B in oral potentially malignant disorders and head and neck carcinomas [39]. These findings seem to contradict the exclusively nuclear A3B localization observed in other tumors [51,52]. Cytoplasmic staining may be attributed to cross-reaction with other APOBEC family members, because the above-mentioned antibodies may cross-react with APOBEC3D (A3D) or APOBEC3A (A3A) which, when endogenously overexpressed, are also localized in the cytoplasm. However, our analysis of HNSCC samples from the UALCAN database showed that A3D, although present, displays an almost 5.5 fold lower expression compared to A3B. Given the fact that the antibody (# ab191695, Abcam) used in this study may cross-react with A3D and taking into consideration the A3D expression analysis in HNSCC, we strongly believe that the majority of the cytoplasmic and nuclear immunostaining can be considered specific and attributed to the presence of A3B but not A3D. Alternative hypotheses for the cytoplasmic localization of A3B include dysfunction of the nucleocytoplasmic transfer machinery in cancer cells; mutations in the nuclear localization signal and nuclear export signal of A3B in cancer cells; or a yet unidentified function of A3B in the cytoplasm [11], other than antiviral activity [53]. Regardless, the role of A3D should be further evaluated in oral and oropharyngeal cancer.

Although it would be preferable to include a large number of patients to acquire more robust results, studies with a small number of participants can be quick to conduct with regard to enrolling patients, reviewing their medical records, performing molecular and biochemical analyses [54]. Therefore, our immunohistochemical results, combined with the analysis of gene expression data from public databases, such as TCGA and UALCAN that include large cohorts of patients, may provide useful insight to form novel hypotheses to be tested in future research.

In OSCC, NF-κΒ expression was associated with bad prognosis [42] and NF-κΒ activation has been associated with a worse prognosis in hematological malignancies and various solid tumors [40]. In contrast, PKCα expression was not associated with tumor grade [44,45] and high expression levels of A3B protein indicate longer disease-free survival rate in patients with breast cancer [49]. In our study, no associations were found between cancer stages, which is considered as a surrogate marker of prognosis, and A3B expression, while there are some indications for an association of PKCα and higher NF-κΒ expression with a less favorable disease stage, an assumption which, nevertheless, requires further verification.

## 5. Conclusions

Our results indicate that the immunohistochemical expression of A3B, a possible source of mutagenesis in non-HPV related oral and oropharyngeal carcinomas, is associated with PKCα/NF-κΒ p65 expression. This suggests that A3B expression in oral and oropharyngeal cancer may be induced through both the classical and the noncanonical NF-κΒ pathway. Additional studies focusing on the activity of A3B, PKCα and NF-κΒ in oral and oropharyngeal carcinomas may reveal their functions in the development and biological behavior of this group of tumors.

## Figures and Tables

**Figure 1 diagnostics-13-00569-f001:**
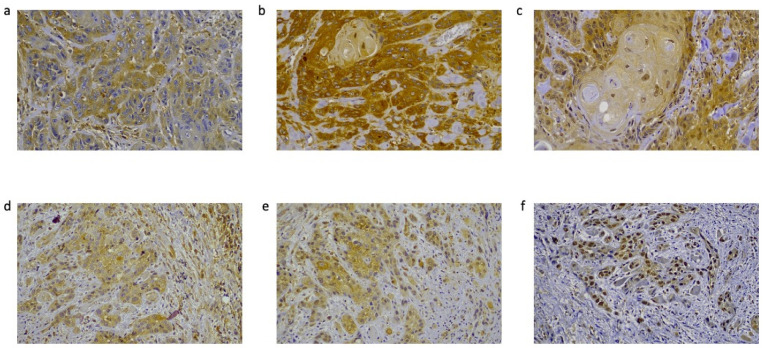
Immunohistochemical expression of PKCα, NF-κΒ, and APOBEC3B: (**a**) cytoplasmic expression of PKCα; (**b**) cytoplasmic expression of NF-κΒ; (**c**) nuclear and cytoplasmic expression of APOBEC3B in an oral squamous cell carcinoma; (**d**) cytoplasmic expression of PKCα; (**e**) cytoplasmic expression of NF-κΒ; and (**f**) nuclear and cytoplasmic expression of APOBEC3B in an oropharyngeal squamous cell carcinoma (initial magnification ×40).

**Figure 2 diagnostics-13-00569-f002:**
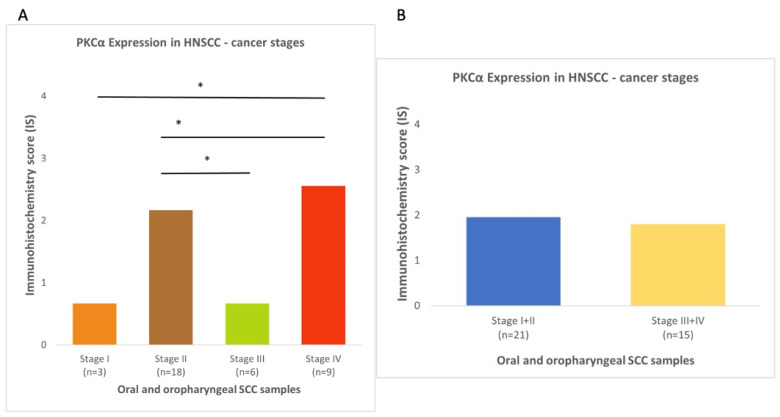
Protein expression of PKCα in oral and oropharyngeal SCC subgroups based on: (**A**) individual cancer stages; (**B**) pooled cancer stages; (**C**) sex; and (**D**) deceased vs. survivors. Protein expression of NF-κΒ in oral and oropharyngeal SCC subgroups based on: (**E**) individual cancer stages; (**F**) pooled cancer stages; (**G**) sex; and (**H**) deceased vs. survivors. Protein expression of APOBEC3B in oral and oropharyngeal SCC subgroups based on: (**I**) individual cancer stages; (**J**) pooled cancer stages; (**K**) sex; and (**L**) deceased vs. survivors. (* *p* < 0.05).

**Figure 3 diagnostics-13-00569-f003:**
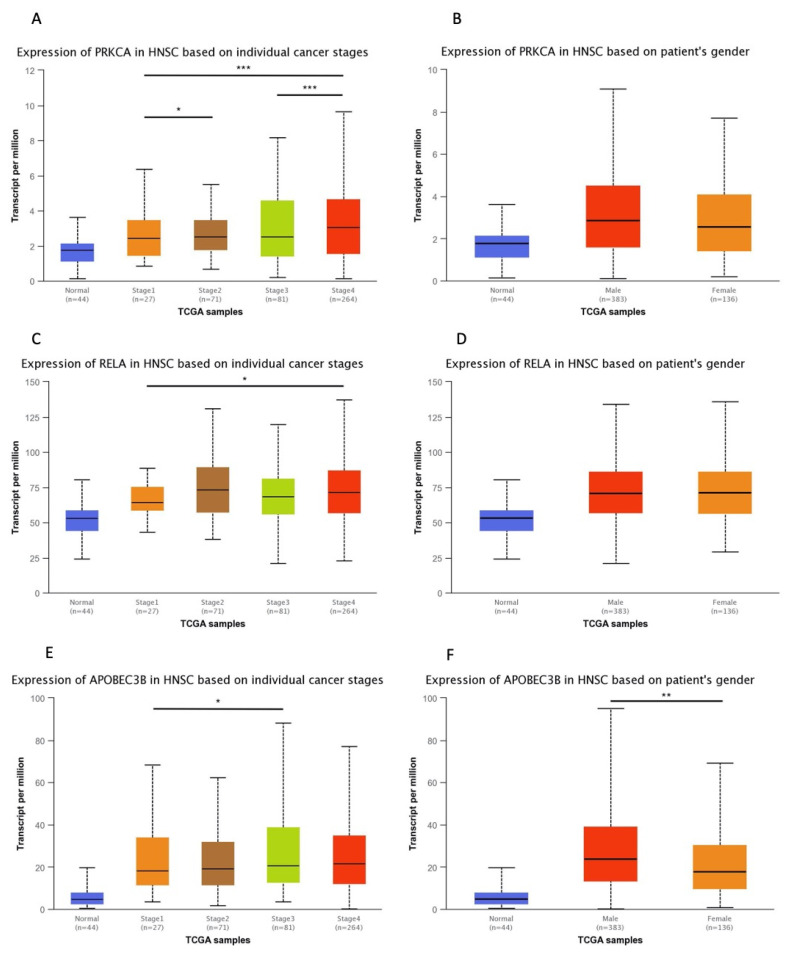
Differential expression analysis of *APOBEC3B*, *PRKCA* and *RELA* mRNA in HNSCC subgroups based on the UALCAN database. mRNA expression of *PRKCA* in HNSCC subgroups based on: (**A**) individual cancer stages; and (**B**) sex. mRNA expression of *RELA* in HNSCC subgroups based on: (**C**) individual cancer stages; and (**D**) sex. mRNA expression of *APOBEC3B* in HNSCC subgroups based on: (**E**) individual cancer stages; and (**F**) sex. (* *p* < 0.05, ** *p* < 0.01, *** *p* < 0.001).

**Table 1 diagnostics-13-00569-t001:** Clinical data.

Number of patients	36
Sex	
Male	24
Female	12
Mean age at diagnosis	63.9
Tumor location	Oral cavity (*n* = 18)
Stage	
I	1
II	8
III	3
IV	6
Tumor location	Oropharynx (*n* = 18)
Stage	
I	2
II	10
III	3
IV	3
Survivors/deceased patients after 60 months	19/14
Patients deceased from other causes	3

**Table 2 diagnostics-13-00569-t002:** PKCa, NF-κΒ and APOBEC3B protein expression in oral and oropharyngeal squamous cell carcinomas.

Oral and Oropharyngeal SCCs *n* = 36	No Staining (0)	Weak—Moderate Staining (1–2)	Strong Staining (4)	Overall Positive
PKCa	7	19	10	29/36 (80%)
NF-κΒ	2	11	22	33/35 (94%)
APOBEC3B	5	13	18	31/36 (86%)

## Data Availability

The data included in this work are available upon request from the corresponding author. Data are not publicly available due to patient privacy. The results shown here are in part based upon data generated by the TCGA Research Network: http://cancergenome.nih.gov/.

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
