# Peer review of "APOBEC3B Is Co-Expressed with PKCα/NF-κB in Oral and Oropharyngeal Squamous Cell Carcinomas"

_diagnostics, 2023, doi:10.3390/diagnostics13030569_

Round 1

Reviewer 1 Report

In the introduction, i suggest the authors to include more recently published articles.

Regarding figure 2 and figure 3  i suggest the authors to make them bigger in order for the data presented to be easier to understand.

Please also specify the limitations of the present study.

Author Response

Response to reviewer's comments

To begin with, we would to thank the reviewer for his/her valuable time and most useful contribution. We do appreciate the inputs given and all comments were of great help in our effort to improve our manuscript.

Point 1: In the introduction, i suggest the authors to include more recently published articles.

Response 1: We have added the following publications, in the introduction section:

  • line 53: PMID: 35859169
  • line 55: PMID: 35918642
  • line 77: PMID: 31954207

Point 2: Regarding figure 2 and figure 3 i suggest the authors to make them bigger in order for the data presented to be easier to understand.

Response 2: Figures 2 and 3 are presented bigger in the revised manuscript, as suggested.

Point 3: Please also specify the limitations of the present study.

Response 3: A paragraph regarding the limitations of the study was included in discussion, as suggested by the reviewer (lines 263-269).

Reviewer 2 Report

This is an interesting study about APOBEC3B co-expression with PKCα/NF-κB in oral and oropharyngeal squamous cell carcinomas.

The paper is well written. However, some issues remain.

Percentages should be added to the results in the abstract and the text.

How was HPV status assessed in the samples? The authors must specifiy if they searched for HPV-DNA or E6/E7 expression or only by means of p16 positivity. The latter is a surrogate for HPV status and it is not enough for a research study.

More data about APOBEC in head and neck cancer from literatura should be added to the Discussion (e.g., PMID 30980272 and 33573337).

Since antibodies for A3B can cross-react with A3A and A3D, the cross-reaction should be excluded before having definitive conclusions in this study.

Author Response

We sincerely thank the reviewer for constructive criticism and valuable comments, which were of great help in revising the manuscript. Accordingly, the revised manuscript has been systematically improved with new information.  

Point 1: Percentages should be added to the results in the abstract and the text.  

Response 1: Percentages were added in the abstract and the results section.  

Point 2: How was HPV status assessed in the samples? The authors must specifiy if they searched for HPV-DNA or E6/E7 expression or only by means of p16 positivity. The latter is a surrogate for HPV status and it is not enough for a research study.

Response 2: All samples were initially evaluated for HPV presence with immunohistochemistry, using a p16 antibody. Further evaluation and confirmation of the original analysis was assessed with DNA in situ hybridization (ISH). Information about the HPV status of all subjects included in this study were retrieved from patients' medical records.

Point 3: More data about APOBEC in head and neck cancer from literatura should be added to the Discussion (e.g., PMID 30980272 and 33573337).

Response 3: Publications regarding APOBEC3B data in head and neck cancer were added to the discussion, as suggested by the reviewer.

  • line 194, reference 33: PMID 30980272
  • line 201, reference 38: PMID 33573337

Point 4: Since antibodies for A3B can cross-react with A3A and A3D, the cross-reaction should be excluded before having definitive conclusions in this study.

Response 4: Indeed, the reviewer's concern on this matter is of great importance and we too, took it under great consideration, during our study design. In this light, we present data (lines 252-258) that support the validity and the APOBEC3B-specific immunostaining results in our study. Moreover, data collected from our studies and from others indicate that cytoplasmic localization of A3B in cancer cells is present in almost all studies and that further evaluation on this matter is necessary for clarification.

Round 2

Reviewer 2 Report

Please report in the text that all samples were initially evaluated for HPV presence with immunohistochemistry, using a p16 antibody and that further evaluation and confirmation of the original analysis was assessed with DNA in situ hybridization (ISH).

Author Response

We appreciate the careful review and constructive suggestions. Accordingly, the revised manuscript has been modified, following the reviewer's comments.

Point 1: Please report in the text that all samples were initially evaluated for HPV presence with immunohistochemistry, using a p16 antibody and that further evaluation and confirmation of the original analysis was assessed with DNA in situ hybridization (ISH).

Response 1: We have added into the text the following "All samples were initially evaluated for HPV presence with immunohistochemistry, using a p16 antibody. Further evaluation and confirmation of the original analysis was assessed with DNA in situ hybridization (ISH)" (lines 90-92), as suggested by the reviewer.